# Non-Enzymatic Glucose Sensing Based on Incorporation of Carbon Nanotube into Zn-Co-S Ball-in-Ball Hollow Sphere

**DOI:** 10.3390/s20154340

**Published:** 2020-08-04

**Authors:** Han-Wei Chang, Chia-Wei Su, Jia-Hao Tian, Yu-Chen Tsai

**Affiliations:** 1Department of Chemical Engineering, National United University, 2, Lienda, Miaoli 36063, Taiwan; hwchang@nuu.edu.tw; 2Department of Chemical Engineering, National Chung Hsing University, 145 Xingda Road, Taichung 402, Taiwan; a70137@smail.nchu.edu.tw (C.-W.S.); zpxoci1950@gms.tku.edu.tw (J.-H.T.)

**Keywords:** Zn-Co-S ball-in-ball hollow nanosphere, carbon nanotube, electrocatalytic behavior, glucose electro-oxidation, electrochemical methods, non-enzymatic glucose sensing

## Abstract

Zn-Co-S ball-in-ball hollow sphere (BHS) was successfully prepared by solvothermal sulfurization method. An efficient strategy to synthesize Zn-Co-S BHS consisted of multilevel structures by controlling the ionic exchange reaction was applied to obtain great performance electrode material. Carbon nanotubes (CNTs) as a conductive agent were uniformly introduced with Zn-Co-S BHS to form Zn-Co-S BHS/CNTs and expedited the considerable electrocatalytic behavior toward glucose electro-oxidation in alkaline medium. In this study, characterization with scanning electron microscopy (SEM), transmission electron microscopy (TEM), X-ray photoelectron spectroscopy (XPS), and X-ray diffraction (XRD) was used for investigating the morphological and physical/chemical properties and further evaluating the feasibility of Zn-Co-S BHS/CNTs in non-enzymatic glucose sensing. Electrochemical methods (cyclic voltammetry (CV) and chronoamperometry (CA)) were performed to investigate the glucose sensing performance of Zn-Co-S BHS/CNTs. The synergistic effect of Faradaic redox couple species of Zn-Co-S BHS and unique conductive network of CNTs exhibited excellent electrochemical catalytic ability towards the glucose electro-oxidation, which revealed linear range from 5 to 100 μM with high sensitivity of 2734.4 μA mM^−1^ cm^−2^, excellent detection limit of 2.98 μM, and great selectivity in the presence of dopamine, uric acid, ascorbic acid, and fructose. Thus, Zn-Co-S BHS/CNTs would be expected to be a promising material for non-enzymatic glucose sensing.

## 1. Introduction

According to the report of global diabetes estimates and projections in 2019 by the International Diabetes Federation (IDF), 463 million people age 20–79 years had diabetes, and the number could increase to 700 million by 2045. The prevalence of diabetes in adults worldwide would increase from 9.3 to 10.9% by 2045 [1]. In Taiwan, diabetes is currently the fifth leading cause of death (data provided by Ministry of Health and Welfare, Taiwan) [2] and the observation shows that diabetes mortality will continue to increase over the coming decades due to unhealthy eating habits and physical inactivity. Type 2 diabetes is the predominant form of diabetes and accounts for 90% of patients diagnosed with diabetes [3]. Thus, the increase of type 2 diabetes may partly reflect increasing clinical diagnosis of diabetes and lead contributor to increasing diabetes prevalence. Therefore, exploring perspectives and practical strategies can be further provided to prevent or delay the prevalence of type 2 diabetes and are effective in reducing the incidence of diabetes. Previous studies showed that the importance of controlling blood glucose variability was an effective strategy to decrease the risk of developing type 2 diabetes. Clearly, there exists a need for the development of fast and efficient approach to determine blood glucose concentration and monitor glucose variability in clinical diagnoses. To compare different detection approaches, it can be assessed that the electrochemical technique may be a choice for performing glucose sensing application due to its low cost, easy fabrication, high reliability, and good reproducibility [4,5]. Currently available electrochemical technique for glucose sensing is enzymatic glucose biosensor and has dominated the market for more than 20 years [6]. However, the most common and serious dilemmas of enzymatic glucose sensor, such as complicated enzyme immobilization process, high cost of enzymes, and critical operational conditions (temperature, pH, humidity and chemical substances), greatly limit their applications of glucose sensing [7]. It is clear that the fabricated non-enzymatic glucose sensing may help overcome these limitations and thereby reflect favorable long-term stability, fast response time and reliable reproducibility toward glucose detection. Recent studies demonstrated that the fabrication of non-enzymatic electrochemical glucose sensing using different electrode materials for detecting glucose possessed excellent detecting performance, including noble metal [8,9], carbon material [10,11], conducting polymer [12,13], and transition metal/metal compound [14,15].

Considerable research on non-enzymatic glucose sensing focused on ternary transition metal oxide/sulfide material with two different transition metal cations that could conduct their excellent catalytic ability originating from multiple oxidation state/structure and enable great potential in glucose sensing [16,17,18,19]. In the transition metal sulfide, S was substituted for O site in the transition metal oxide and might produce a more flexible structure originating to the lower electronegativity of S (the electronegativity of S (2.5) was smaller than that of O (3.5)), which led to relatively good electrical conductivity and narrow optical band gap [20,21]. Recently, controlling the ionic exchange reaction to synthesize ternary transition metal sulfide ball-in-ball hollow sphere (BHS) was applied as an efficient strategy to develop high performance electrode materials consisting of multilevel structures, which opened up new opportunities for varied applications, resulting in further improvement of the electrochemical performance [22,23]. Being one of the ternary transition metal sulfide, Zn-Co based mixed sulfide demonstrated excellent electrochemical properties and rendered it accessible for promising application in the field of electrochemistry, including supercapacitors [24,25], water splitting [26,27], and rechargeable battery [28,29]. These features made Zn-Co based mixed sulfide as a potentially ideal material for non-enzymatic glucose sensing. Previous studies also reported that the incorporation of conductive carbon material into transition metal oxide/sulfide material could further reduce the pulverization of active material, displaying the excellent rate capability, cycling stability, and electronic conductivity during electrochemical process [30,31,32].

In this work, Zn-Co-S BHS was successfully prepared by solvothermal sulfurization method. Further, the fabrication of Zn-Co-S BHS/CNTs by the incorporation of conductive CNTs into Zn-Co-S BHS to enhance the electrical conductivity and reduce the pulverization of active materials proposed a reliable strategy to further improve the glucose sensing performance. It was expected that Zn-Co-S BHS/CNTs could be a promising electrode material for non-enzymatic electrochemical glucose sensing.

## 2. Materials and Methods

### 2.1. Reagents

Zinc nitrate hexa-hydrate (Zn(NO_3_)_2_·6H_2_O), cobalt nitrate hexa-hydrate (Co(NO_3_)_2_·6H_2_O), thioacetamide (TAA) were obtained from Alfa Aesar (Haverhill, MA, USA). d-(+)-glucose, uric acid (UA), dopamine (DA), ascorbic acid (AA), fructose (FR), sodium hydroxide were purchased from Sigma-Aldrich(St. Louis, MO, USA). Isopropyl alcohol, glycerol, anhydrous ethanol (C_2_H_5_OH, 99.5%) were purchased from J.T. Baker. Carbon nanotubes (CNTs) were obtained from Golden Innovation Business Co., Ltd (Taipei, Taiwan). All water used was deionized water (DI water) through Milli-Q water purification system (Millipore, MAs, USA). All chemicals were used without further purification.

### 2.2. Preparation of Zn-Co-S BHS/CNTs

Zn-Co-S BHS was successfully synthesized by using Zn/Co bimetallic precursor (Denoted as ZnCo-precursor) through a hydrothermal sulfurization process. The ZnCo-precursor was obtained by a facile hydrothermal method. First, 0.25 mmole Zn(NO_3_)_2_·6H_2_O, 0.5 mmole Co(NO_3_)_2_·6H_2_O, and 7.5 mL glycerol were dissolved in the 52.5 mL isopropanol under ultrasonic treatment for 30 min to get transparent pink solution and then transferred into a 100 mL Teflon container in a stainless steel autoclave to react at 180 °C for 24 h. The resulting Zn-Co precursor were separated by centrifugation, and washed with ethanol and DI water for several times and dried in an oven at 60 °C. Second, 30 mg of Zn-Co precursor and 50 mg TAA were dissolved into 20 mL of ethanol, and the sulfurization process was reacted at 190 °C for 12 h. After cooled down at room temperature, the black precipitate of Zn-Co-S BHS was obtained, washed with ethanol and DI water for two times, and dried in oven at 60 °C overnight. The incorporation of conductive CNTs into Zn-Co-S BHS was prepared as follows: 1 mg CNTs and 2 mg Zn-Co-S BHS were dispersed in 1.0 mL of isopropanol. The formation of the resulting material was designated as Zn-Co-S BHS/CNTs. Subsequently, 6 μL of Zn-Co-S BHS/CNTs was dropped on the surface of the glassy carbon electrode (GCE, diameter: 3 mm), and then 4.5 μL 0.5 wt% Nafion uniformly cover on the surface of Zn-Co-S BHS/CNTs electrode, allowed to dry at room temperature, and collected for subsequent treatment.

### 2.3. Apparatus

The morphological and physical/chemical properties were characterized by using scanning electron microscopy (SEM, JSM-7410F, JEOL, Tokyo, Japan), transmission electron microscopy (TEM, JEM-2100F, JEOL, Tokyo, Japan), X-ray photoelectron spectroscopy (XPS, PHI-5000 Versaprobe, ULVAC-PHI, Chigasaki, Japan), and X-ray diffraction (XRD, D8 Discover SSS, Bruker D & Advance, Bruker, Germany). Electrochemical measurements were performed by electrochemical analyzer (Autolab, model PGSTAT30, Eco Chemie, Utrecht, Netherlands). The electrochemical cell with three-electrode systems were comprised of as-prepared samples-modified glassy carbon working electrode, a platinum wire counter electrode, and an Ag/AgCl (3 M KCl) reference electrode. The cyclic voltammogram (CV) was recorded in the voltage range 0~+0.7 V (versus Ag/AgCl) in 0.1 M NaOH in the absence and presence of 1.0 mM glucose at a scan rate of 20 mV s^−1^. And chronoamperometric current-time detection of glucose was obtained in 0.1 M NaOH upon successive addition of glucose with a time interval of 20 s at applying voltage +0.6 V. Interference and real sample analysis were evaluated at applying voltage +0.6 V using the chronoamperometry (CA) with additions of 25 M interfering biomolecules (including dopamine (DA), ascorbic acid (AA), uric acid (UA) and fructose (FR)) and synthetic serum.

## 3. Results and Discussions

The morphology of Zn-Co-S BHS was characterized by SEM and TEM. SEM images (Figure 1a,b) revealed that the Zn-Co-S BHS with an average size of about 500 nm exhibited a rough surface owing to its unique structural characterization, which endowed it with a large specific surface area. An individual Zn-Co-S BHS was displayed in the enlarged TEM image (Figure 1c). TEM image clearly showed that Zn-Co-S BHS constructed ball-in-ball-like hollow sphere through anion exchange reaction during the sulfurization process between the anion diffusing inwards and cation diffusing outwards, which would further prompt the growth of the Zn-Co-S shell and form the well-defined structural feature [33]. Zn-Co-S BHS showed an average shell thickness of about 25 nm, and the obtained Zn-Co-S shell was composed of a much smaller primary particle with relatively high specific surface area. Zn-Co-S BHS with large surface area afforded an increased electrolyte/electrode contact area and hence provided more surface active sites that were expected to favor the efficient transfer of electron/ion in glucose electro-oxidation and further improve glucose sensing performance.

The structure and phase composition of Zn-Co-S BHS were characterized by XRD pattern in comparison with the standard patterns of Zn_0.76_Co_0.24_S (JCPDS No.47-1656) and CoS (JCPDS No.75-0605), as shown in Figure 2. The XRD pattern indicated that Zn-Co-S BHS exhibited several distinct diffraction peaks at 2θ about 28.66°, 30.5°, 35.2°, 47.68°, 54.2°, 56.34°, and 77.06°, which were very close to the standard diffraction patterns of Zn_0.76_Co_0.24_S and CoS. The surface elemental composition and valance state of Zn-Co-S BHS were characterized by XPS spectra, as showed in Figure 3. Recorded survey spectrum of Zn-Co-S BHS revealed that Zn, Co and S were the main elements of Zn-Co-S BHS (Figure 3a). The Zn 2p XPS spectrum (Figure 3b) showed that the spin-orbit splitting of the two peaks, the spin-orbit split doublet with separation between peaks of Zn 2p_3/2_ and Zn 2p_1/2_ at 1021.2 eV and 1044.3 eV, was about 23 eV, which indicated the presence of Zn oxidation state in the Zn-Co-S BHS [34]. Figure 3c showed the peaks located at 779.8 eV in Co 2p_3/2_ and 796.4 eV in Co 2p_1/2_ in the Co 2p XPS spectrum. The spin-orbit splitting of the two peaks was about over 15 eV, indicating the co-existence of Co^2+^ and Co^3+^ in the Zn-Co-S BHS [35]. The S 2p XPS spectrum was shown in Figure 3d, which was known to contain the S 2p_3/2_ and the S 2p_1/2_ spin-orbit splitting at 161.1 eV and 162.4 eV and one shake-up satellite (Sat.) [36]. Therefore, the results of above analyses demonstrated that Zn-Co-S BHS was successfully prepared and exhibited excellent morphological and structural property. For practical applications, the incorporation of conductive CNTs into Zn-Co-S BHS forming a conductive network between Zn-Co-S BHS and CNTs could bridge effective interconnected conductive network, which offered great advantage for constructing non-enzymatic glucose sensing.

Electrochemical analyses were performed by cyclic voltammetry (CV) and chronoamperometry (CA). Figure 4 showed the CV curves of CNTs, Zn-Co-S BHS, and Zn-Co-S BHS/CNTs in 0.1 M NaOH in the absence (dashed lines) and presence (solid lines) of 1.0 mM glucose at a scan rate of 20 mV s^−1^. In Figure 4b,c, Zn-Co-S BHS and Zn-Co-S BHS/CNTs had a pair of redox peaks at around 0.2~0.4 V (marked by a blue star). However, no redox peaks was observed at CNTs (Figure 4a). This result indicated that redox peaks in the Zn-Co-S BHS and Zn-Co-S BHS/CNTs should be attributed to the existence of Faradaic redox couple species in Zn-Co-S BHS, which might act as active sites to deliver enhanced electrochemical activity during glucose electro-oxidation [37]. By comparing the CV curves of Zn-Co-S BHN/CNTs with CNTs and Zn-Co-S BHS in 0.1 M NaOH, Zn-Co-S BHS/CNTs exhibited satisfactory electrochemical catalytic activity, which was proposed to derive from the unique conductive network of CNTs providing an ideal conductive contact between Zn-Co-S BHS and CNTs. It was expected to offer a highway to facilitate electron/ion transferring into electroactive sites within Zn-Co-S BHS, supporting an effective strategy to further boost the performance of non-enzymatic glucose sensing. Upon adding 1.0 mM glucose, anodic oxidation peak appeared at about 0.5~0.6 V corresponding the onset of glucose electro-oxidation. The comparisons showed that Zn-Co-S BHS/CNTs exhibited the relatively large changes in anodic oxidation peak current before and after the glucose addition (marked by a blue arrow), which indicated that the synergistic effect of Faradaic redox couple species of Zn-Co-S BHS and unique conductive network of CNTs exhibited excellent electrochemical catalytic ability towards the glucose electro-oxidation.

In order to achieve better performance for glucose electro-oxidation, applied voltage and the proportion of Zn-Co-S BHS/CNTs were carried out to have more insight into the optimization of the operating parameters affecting electrochemical catalytic performance. Zn-Co-S BHS/CNTs with different applied voltages and the proportions of Zn-Co-S BHS/CNTs could be characterized by CA in 1 M NaOH and 1.0 mM glucose. CA curves for glucose electro-oxidation with different applied voltage from 0.5 to 0.65 V were displayed in Figure 5a. With the addition of 1.0 mM glucose, the current response gradually increased and tended to quite stable. The current response increased significantly with increasing applied voltage and reached a maximum at applied voltage +0.6 V. However, further increase in applied voltage resulted in the decrease of current response and became unstable because high applied voltage could lead to the fluctuation and instability with the generation of intermediates on the surface of Zn-Co-S BHS/CNTs. Therefore, +0.6 V was selected as the best applied voltage with the glucose electro-oxidation. Figure 5b displayed the CA for CNTs (1 mg) and different weight of the Zn-Co-S BHS of 1 mg, 1.5 mg, 2 mg, and 2.5 mg by applying voltage +0.6 V. By contrast, the Zn-Co-S BHS/CNTs (2 mg/1 mg) exhibited dramatic increases in current response. This could be explained by the fact that Zn-Co-S BHS/CNTs with 2 mg/1 mg proportion were very well dispersed in the isopropanol, which were very suitable to reach a suitable conductivity and form more effective conductive network by suitable addition ratio of Zn-Co-S BHS and CNTs, and facilitated rapid electron/ion transportation. Therefore, the optimization of great dispersed Zn-Co-S BHS/CNTs with 2 mg/1 mg proportion and applied voltage +0.6 V played in defining electrochemical catalytic reforming and demonstrated a significant enhancement of electrocatalytic performance for the non-enzymatic electro-oxidation of glucose.

After the optimization, Zn-Co-S BHS/CNTs was used for fabricating a non-enzymatic amperometric glucose sensing to analyze the performance towards the glucose electro-oxidation. Figure 6a showed a typical amperometric current-time plot of Zn-Co-S BHS/CNTs in 0.1 M NaOH with successive addition of distinct glucose concentration at applied voltage +0.6 V. The Zn-Co-S BHS/CNTs exhibited a rapid and sensitive current response generated after each addition of glucose concentration. With the increase in glucose concentration, the current response also increased and finally reached saturation at high-glucose concentration. The calibration curve was presented in Figure 6b, exhibiting the linear range of the calibration curve between 5 to 100 μM with sensitivity of 2734.4 μA mM^−1^ cm^−2^, and the linear equation was I = 0.191 Conc. +1.026 with a correlation coefficient of 0.993. The detection limit was estimated 2.98 μM (S/N = 3) (S was the standard deviation of the blank signals for *n* = 10, and N was the slope of the calibration curve). The sensor behaviors of Zn-Co-S BHS/CNTs were better or comparable to the present reported non-enzymatic glucose sensing based on ternary transition metal oxide/sulfide based materials (Table 1) [37,38,39,40,41,42].

Many interfering biomolecules, dopamine (DA), ascorbic acid (AA), uric acid (UA) and fructose (FR) normally co-exist with glucose in human blood. Thus, the interference experiment was performed to evaluate the selectivity of Zn-Co-S BHS/CNTs with successively injecting 50 μM glucose, 25 μM DA, 25 μM AA, 25 μM UA, 25 μM FR, and 50 μM glucose into 0.1 M NaOH solution at applied voltage +0.6 V. Figure 7 showed that the Zn-Co-S BHS/CNTs exhibited a high selectivity for glucose and did not cause observable interference signal. The result suggested that Zn-Co-S BHS/CNTs were highly specific to glucose, even detecting glucose in the presence of the common interferences. However, it was not clear how the selective electrocatalytic sensing of interfering biomolecules is in the presence of glucose. Present evidence suggested that it was probably due to the synergistic effect of geometry configuration and isoelectric point of the selected electrode materials, resulting in an improved selectivity [42,43]. For real sample analyses, Zn-Co-S BHS/CNTs were applied to the determination of glucose concentration in synthetic serum. The synthetic serum was a saline buffer consisting of 0.05 M PBS, 0.1 M KCl, 4.83 mM glucose, 0.05 mM ascorbic acid, 0.5 mM uric acid, 1.05 mM lactic acid, 5 mM urea, 60 μM Pyruvic acid, 35 μM Cysteine, 0.50 mM bovine albumin, and 3.96 mM cholesterol. The results for real sample analyses were measured good recovery (~98.6%) and repeatability (RSD = 5.1%) were obtained. The glucose sensing fabricated by Zn-Co-S BHS/CNTs offered a reliable recovery of glucose detection in synthetic serum and presented new opportunities for their practical application.

## 4. Conclusions

In this study, Zn-Co-S BHS was successfully synthesized by solvothermal sulfurization method. Subsequently, the incorporation of conductive CNTs into Zn-Co-S BHS formed a conductive contact between Zn-Co-S BHN and CNTs to provide efficient electron/ion conductive pathway for glucose electro-oxidation. The electrochemical catalytic performance through optimizing the applied voltages (+0.6 V) and the proportion of Zn-Co-S BHS/CNTs (2 mg/1 mg) would be supplied to demonstrate the effect of the electrochemical promotion. The excellent electrochemical catalytic ability might be ascribed to the synergistic effect of multiple redox couple species of Zn-Co-S BHS and unique conductive network of the CNTs towards the glucose electro-oxidation. Zn-Co-S BHS/CNTs exhibited high response sensitivity (2734.4 μA mM^−1^ cm^−2^), great detection limit (2.98 μM), and good selectivity. The remarkable performance of Zn-Co-S BHS/CNTs provided the opportunity as a non-enzymatic electrochemical glucose sensing for point-of-care diagnosis, food analysis, and biotechnology.

## Figures and Tables

**Figure 1 sensors-20-04340-f001:**
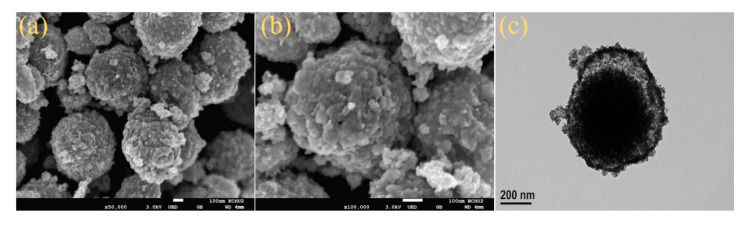
(**a**,**b**) SEM and (**c**) TEM images of Zn-Co-S BHS.

**Figure 2 sensors-20-04340-f002:**
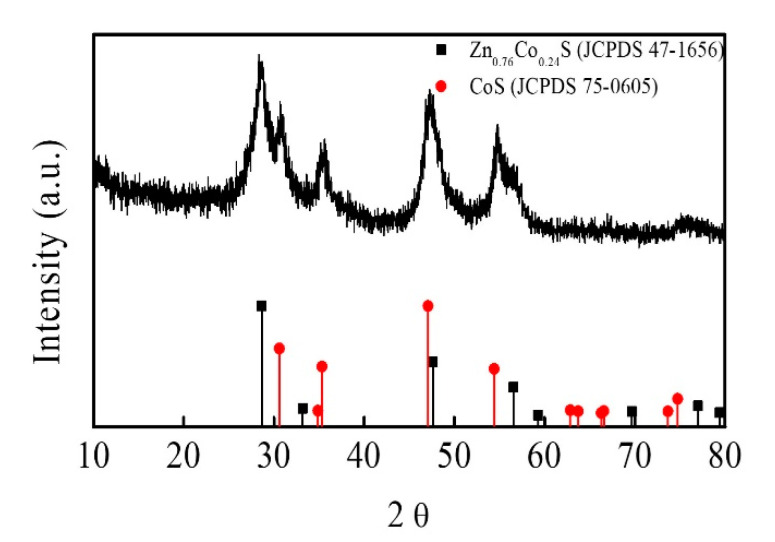
XRD patterns of Zn-Co-S BHS.

**Figure 3 sensors-20-04340-f003:**
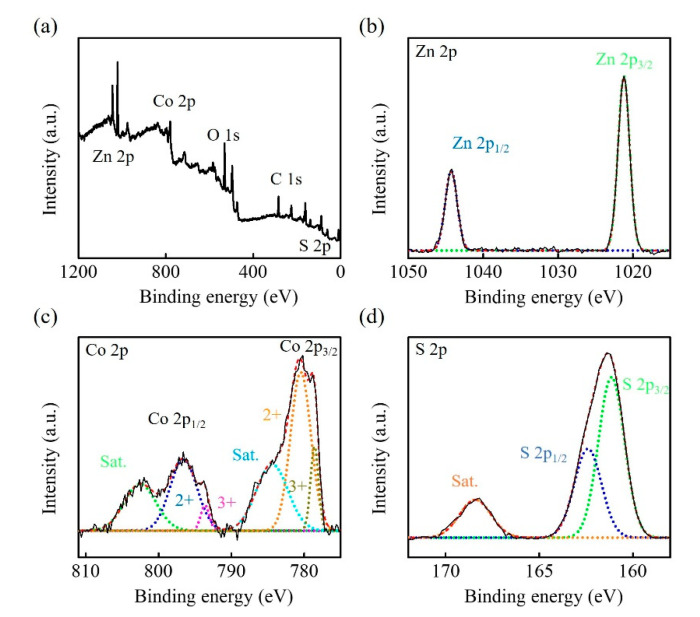
XPS spectra of Zn-Co-S BHS (**a**) full scan, (**b**) Zn 2p, (**c**) Co 2p, and (**d**) S 2p.

**Figure 4 sensors-20-04340-f004:**
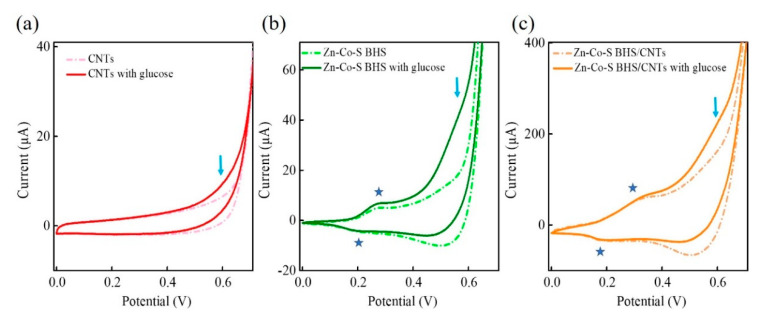
CV curves of (**a**) CNTs, (**b**) Zn-Co-S BHS, and (**c**) Zn-Co-S BHS/CNTs in 0.1 M NaOH in the absence (dashed lines) and presence (solid lines) of 1.0 mM glucose at a scan rate of 20 mV s^−1^.

**Figure 5 sensors-20-04340-f005:**
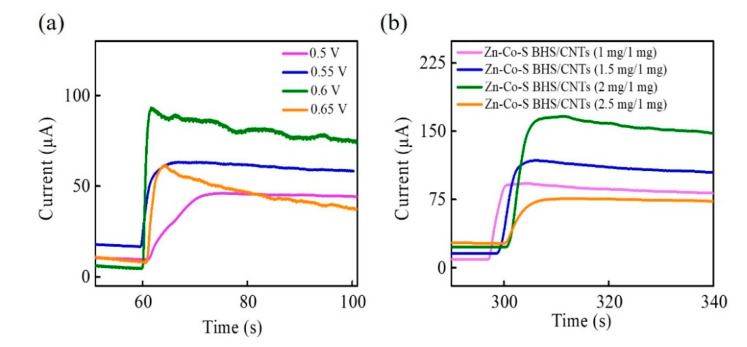
CA curves of Zn-Co-S BHS/CNTs with (**a**) different applied voltages and (**b**) CNTs (1 mg) and different the weight of Zn-Co-S BHS in 0.1 M NaOH in the 1.0 mM glucose.

**Figure 6 sensors-20-04340-f006:**
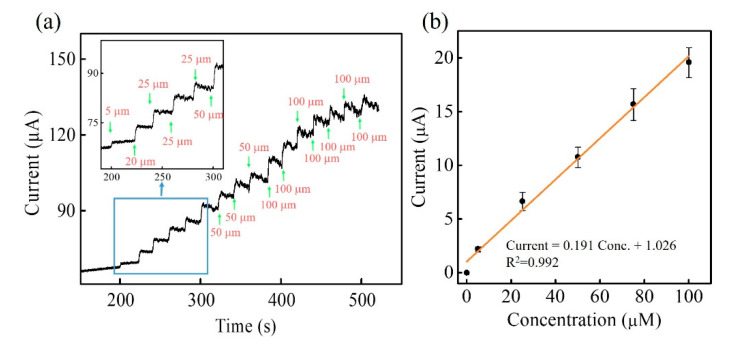
(**a**) Current-time plots and (**b**) calibration curve of Zn-Co-S BHS/CNTs in 0.1 M NaOH with successive addition of various glucose concentration at applying voltage +0.6 V. In inset of Figure 6a: the linear range of Zn-Co-S BHS/CNTs from 5 to 100 μM.

**Figure 7 sensors-20-04340-f007:**
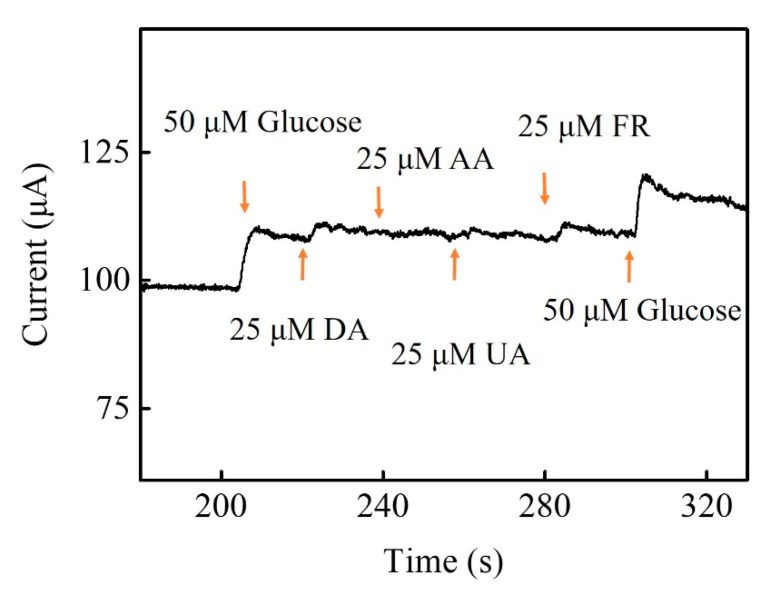
Interference tests of Zn-Co-S BHS/CNTs.

**Table 1 sensors-20-04340-t001:** Performance comparison of non-enzymatic glucose sensing based on ternary transition metal oxide/sulfide materials.

Type of Materials	Linear Range	Sensitivity (μA mM^−1^ cm^−2^)	Detection Limit	Reference
Zn-Co-S BHS/CNTs	5~10 μM	2734.4	2.98 μM	This work
NiCo_2_S_4_/Ni/cellulose filter paper	0.5 μM~6 mM	283	50 nM	[37]
NiCo_2_O_4_/carbon nanofiber	5 μM~19.175 mM	1947.2	1.5 μM	[38]
NiCo_2_S_4_	5 μM~0.1 mM 0.25~2 mM	858.57 332.84	2 μM	[23]
NiCo_2_O_4_	0.3 μM~1 mM	1685.1	0.16 μM	[39]
Co_3_O_4_/ NiCo_2_O_4_/graphene	0.01~3.52 mM	304	0.384 μM	[40]
NiCo_2_S_4_	0.2~2.4 mM	1890	2.23 μM	[41]

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
