# Peer review of "Non-Enzymatic Glucose Sensing Based on Incorporation of Carbon Nanotube into Zn-Co-S Ball-in-Ball Hollow Sphere"

_sensors, 2020, doi:10.3390/s20154340_

Round 1
Reviewer 1 Report
The manuscript subject by Chia-Wei Su, Jia-Hao Tian, ​​Han-Wei Chang and Yu-Chen Tsai "Non-enzymatic glucose sensing based on incorporation of carbon nanotube into Zn-Co-S ball-in-ball hollow nanosphere" is relevant. A new version of the nanocomposite for sensitive electrochemical determination of glucose is proposed. Developed sensors can be important in terms of their practical application. At the same time, there are a number of significant issues that need to be solved for the practical use of such sensors in medical diagnostics - this is the stability of the sensors and their selectivity. In this regard, the following questions arise:
- The sensor is made by simply applying a composite on the surface of the glassy carbon electrode followed by applying Nafion. How firmly is the composite held onto the electrode surface? How long does it keep stability? In what form is it planned to implement this sensor for practical use? This is a very significant issue, as there are many research articles on this topic, it is necessary to describe the obvious advantages in terms of the practical implementation of the proposed sensors.
- It is known that biomolecules such as dopamine, ascorbic acid and uric acid are significantly easier to oxidize compared to glucose, even in the absence of catalysis. How to explain the “phenomenon" of the oxidation effect absence of these components on the total oxidation current?
- The authors write: «Figure 4 clearly showed that Zn-Co-S BHN and Zn-Co-S BHN/CNT had redox peaks at around 0.25 ~ 0.30 V, and the redox peaks might be related to Zn2+/Zn3+ and Co2+/Co3+ 170 reversible redox reaction in the Faradaic process due to the existence of Zn-Co-S BHN.» This is not true. According to the figure, we see a weakly expressed oxidation signal and only in the case of the Zn-Co-S BHN / CNT. It is impossible to draw conclusions about the severity of signals, and even more so about the reversibility of processes from such data. I would like to see more substantial evidence of the catalytic abilities of this nanocomposite.
Reviewer 2 Report
This article describes the development of a glucose sensor based on non-enzymatic means. They have used a Zn-Co material that is commonly used in water splitting and battery research and showed its capability as an electrode material for glucose sensing. The material is very well characterised. However in saying that the sensing experiments are not clear. The experimental section is very bare for the sensing aspect. Furthermore, in the results and discussion section more explanation is needed for the selectivity and recovery experiments. Figures are also really hard to look at.
The immediate introduction of Zn-Co-S ball in ball hollow nanosphere(BHN) in the abstract should be looked at and a clearer abstract is recommended. What is BHN why use it? What is the focus of the work?
EDX analysis with the SEM of the particles would have been really helpful in the characterisation step and show the shelled structure clearer then TEM.
Line 75: BHN is mentioned in the introduction without explaining what it is.
It was mentioned that the material was dropped on the surface of a glassy carbon electrode for measurements. How good was the attachment? Was it a uniform film? Was there any coffee stain effect?
The size of the particle is relatively high to be considered a nanocomposite.. 500 nm is more half a micron sized particle.
Why is the material particularly selective to glucose? That has not been discussed in the results section.
Figures 3, 4, 5 and 6 need to be bigger. It is very hard to see the results.
References for claims in lines 173-175?
In line 193 it was mentioned that the oxidation peak is relatively large. This is not really clear in the figure. There is a relatively large capacitance of the Zn-Co-S BHN/CNT materials as seen in figure 4. What is the reason for this? This large capacitance is affecting the redox peaks seen at 0.5-0.6V. A more sensitive technique such as DPV or SWV might clearly show the analytical signal.
In line 215 it is mentioned that figure 5b displays the CV. It should be CA.
Line 234: How was different concentrations of glucose achieved? Was the solution spiked with a stock solution? What was the time interval? How long was saturation? The later concentrations in figure 6a are barely visible.
How was recovery and selectivity experiments performed in figure 7. This needs more expanding. Lines 256 to 264. There is no mention of how sensing experiments were performed even in the methods section.
Round 2
Reviewer 1 Report
Answers and explanations accepted
Author Response
Thanks for your suggestions.
Reviewer 2 Report
The authors have taken on board all the suggestions and comments and made significant changes to the paper.
Author Response
Thanks for your suggestions.